# Practical Implications of Different Phenotypic and Molecular Responses of Evergreen Conifer and Broadleaf Deciduous Forest Tree Species to Regulated Water Deficit in a Container Nursery

**Piotr Robakowski** [1,*], **Tomasz P. Wyka** [2], **Wojciech Kowalkowski** [1], **Władysław Barzdajn** [1], **Emilia Pers-Kamczyc** [3], **Artur Jankowski** [2,3] **and Barbara Politycka** [4]

[1] Faculty of Forestry, Poznan University of Life Sciences, Wojska Polskiego 71E, 60-625 Poznań, Poland; wojciech.kowalkowski@up.poznan.pl (W.K.); barzdajn@up.poznan.pl (W.B.)
[2] General Botany Laboratory, Faculty of Biology, Adam Mickiewicz University, Uniwersytetu Poznańskiego 6, 61-614 Poznań, Poland; twyka@amu.edu.pl (T.P.W.); artur.jankowski@amu.edu.pl (A.J.)
[3] Institute of Dendrology, Polish Academy of Sciences, Parkowa 5, 62-035 Kórnik, Poland; epk@man.poznan.pl
[4] Department of Plant Physiology, Faculty of Horticulture and Landscape Architecture, Poznan University of Life Sciences, Wołyńska 35, 60-625 Poznań, Poland; barbara.politycka@up.poznan.pl
* Correspondence: piotr.robakowski@up.poznan.pl; Tel.: +48-61-848-77-38

**Abstract:** Recent climatic changes have resulted in an increased frequency and prolonged periods of drought and strained water resources affecting plant production. We explored the possibility of reducing irrigation in a container nursery and studied the growth responses of seedlings of four economically important forest trees: broadleaf deciduous angiosperms *Fagus sylvatica* L., *Quercus petraea* (Matt.) Liebl., and evergreen conifers *Abies alba* Mill. and *Pinus sylvestris* L. We also studied markers of water stress including modifications of biomass allocation, leaf anatomy, proline accumulation, and expression of selected genes. Growth of the broadleaved deciduous species was more sensitive to the reduced water supply than that of conifers. Remarkably, growth of the shade tolerant *Abies* was not affected. Adjustment of biomass allocations was strongest in *P. sylvestris*, with a remarkable increase in allocation to roots. In response to water deficit both deciduous species accumulated proline in leaves and produced leaves with shorter palisade cells, reduced vascular tissues, and smaller conduit diameters. These responses did not occur in conifers. Relative transcript abundance of a gene encoding the Zn-finger protein in *Q. petraea* and a gene encoding the pore calcium channel protein 1 in *A. alba* increased as water deficit increased. Our study shows major differences between functional groups in response to irrigation, with seedlings of evergreen conifers having higher tolerance than the deciduous species. This suggests that major water savings could be achieved by adjusting irrigation regime to functional group or species requirements.

**Keywords:** biomass allocation; drought; irrigation; leaf anatomy; mRNA level; proline

## 1. Introduction

According to predictions, global warming will lead to an increased frequency and extent of summer drought periods in many regions of the Earth [1]. Although in northern and western European forests global warming is expected to stimulate forest growth, the accompanying drought may cause adverse effects [2]. This is because climate warming will increase evapotranspiration more than precipitation and periodic drought events will enhance effects of increased temperature on growth and photosynthesis of boreal and temperate tree species [3,4]. The understanding of drought effects on trees is thus vital for proper management and conservation of forest ecosystems.

Water shortages affect all biological functions of trees, with the risk of drought-induced mortality depending on the local geographical conditions, precipitation, soil properties and anthropogenic factors [5]. Vulnerabilities of individual species vary widely [6–9]. Much knowledge of the variability of tree responses to drought has been gathered from studies of mature stands [10–12], however, the most vulnerable stage in the tree's ontogeny is the seedling. In particular, it is not certain to what extent the responses and vulnerabilities of adult trees vs. seedlings to drought are consistent in between-species comparisons [13,14].

Trees acclimate to restriction in water supply through modifications of growth, biomass allocation, organ structure, and through functional adjustments encompassing processes from gene expression to enzymatic activity and metabolic regulation [7,15–17]. These adaptive changes may lead to improved plant survival through an increased water uptake and a more conservative water use [18]. Water deficit typically leads to an increase in relative biomass allocation to roots, a decrease in allocation to leaves, and a reduced leaf area ratio and leaf mass fraction, resulting in greater water uptake potential and reduced transpiration [19–21]. Individual leaves produced under the influence of drought are usually smaller than those from well hydrated plants [22–24] and show reduced amount of vascular tissues and smaller diameters of conductive elements [25–27]. Plants subjected to drought may accumulate metabolites responsible for osmotic adjustment and oxidative stress protection as well as change hormone and regulatory molecule concentrations [28–30]. Profiles of gene expression and enzymatic as well as transport activities may become greatly altered [31,32]. Adjustments occur at different time scales, but ultimately may result in drought resistant or drought tolerant phenotypes. The ability to produce such plastic modification may be the key to plant survival under natural settings. However, traits that enhance drought survival, especially those related to water conservation, frequently trade off with growth potential [33]. For example, increased allocation to roots may lead to a decreased ratio of assimilatory to non-assimilatory biomass, restricting whole-plant carbon supply even after the cessation of drought [34]. Moreover, long term carry-over effects (so called drought memory) may also result from metabolic adjustment to drought [35,36].

Water requirements and responses to drought differ significantly among taxa and functional groups. For major forest tree species, the grouping into drought tolerance categories or rankings has been based on characteristics of their habitat and responses of growth and physiological indices to environmental drought [6,7]. In general, tree species adapted to drier environments are more tolerant to drought compared with those from humid habitats [37,38]. At the functional group level, evergreen conifers are more tolerant to drought than deciduous broadleaved tree species [6] and late successional, shade tolerant tree species are often more sensitive to water deficit than pioneer and shade intolerant species [39]. Many insights have been provided by physiological studies, especially those focusing on vulnerability to cavitation and responses of stomatal conductance [7,40]. Such results may provide general information on species requirements and drought vulnerability ranking, yet, since they focus on saplings or adult trees, their relevance for seedling stages may be limited.

The species selected for this study were two deciduous broadleaf angiosperms (*Fagus sylvatica* L. and *Quercus petraea* (Matt.) Liebl. and two evergreen conifers (*Abies alba* Mill. and *Pinus sylvestris* L.). They differ in ecological requirements with respect to moisture and their overall sensitivity to water deficit has been investigated under various settings. Briefly, *Fagus sylvatica* is a dominant European forest tree species of high economic and ecological importance. It is a shade-tolerant, late successional, and the most water-demanding tree among our selected species [7,41]. It requires high air humidity and grows preferentially on fertile, well-drained and moist, but not waterlogged, soils. *Quercus petraea* is a moderately shade tolerant, late successional tree. It maintains higher leaf conductance [42] and higher net $CO_2$ assimilation rates than *F. sylvatica* under the same experimental conditions [43,44]. *Abies alba* is a strongly shade tolerant, late successional, and slow-growing tree [45]. It is considered rather sensitive to drought [46,47]. *Pinus sylvestris* is the most widely distributed Eurasian conifer. It is a pioneer, early-successional, light-demanding, and drought-tolerant tree. Severe drought, however, has a limiting effect on *P. sylvestris* growth and may trigger Scots pine decline [48]. It may be expected

that species' growth sensitivity to low water supply should follow ecological species ranking [41]. Our study species can be classified according to their decreasing drought tolerance in the following order: *P. sylvestris* > *A. alba* > *Q. petraea* > *F. sylvatica* [49]. However, this ranking cannot be generalized to seedlings and to the whole geographical ranges of these species. In addition, it is not clear whether seedlings of our study species growing in a container nursery will show similar drought tolerance as in natural conditions.

Raising seedlings under polyethylene tunnel has become an important method of containerized forest tree production [50]. Usually, seedlings are grown in containers placed under an unheated tunnel. Under Central European conditions, seed planting takes place in May and seedlings may be grown until the end of July, August, or September, depending on the species, and then the seedlings are transferred to an outdoor nursery. Nutrients are provided with water used for irrigation or as slow-release fertilizers added to the substrate. Irrigation in container forest nurseries is usually carried out by programmable automatic systems delivering water according to established norms [51]. The rising cost of water provides incentive to reduce water consumption, however decreasing irrigation doses below physiological optimum may result in water stress and decrease the development rate of seedlings. On the other hand, reduced substrate water content may improve substrate aeration which is beneficial for growth and the quality of seedlings [52] and reduces nutrient leaching from the substrate [53]. Lower substrate water content, especially at the end of the growing season, accelerates tissue maturation and bud set, resulting in improved frost tolerance [54]. These additional benefits of reduced irrigation thus occur in addition to water savings.

In this study we evaluated the effects of reduced irrigation on growth and on several morphological, anatomical, metabolic, and molecular variables that are known to predictably respond to water deficit. We studied whole plant allocation of biomass among major organ types (leaves, roots, and stems), examined some key anatomical dimensions of foliar cross-section, and determined the concentration of proline—a compound involved in the process of osmotic adjustment [55,56]. Additionally, we studied the expression of genes known to respond to water deficit in our focal species. Although the molecular basis of phenotypic responses of forest trees to water deficit is not well understood, and the identification of water deficit related genes in trees, especially in conifers, has been difficult, partly because of their large genomes [57], sets of such reference drought-responsive genes have been identified first in *Arabidopsis* and some crops [58], and later in other plants, including trees [59,60]. These genes may be considered as indicators of drought stress [61].

We tested the hypotheses that under decreased irrigation water supply: (1) seedling growth will be reduced and the reduction of biomass accumulation will be relatively larger in broadleaved deciduous and shade-tolerant species than in conifers and light-demanding species; (2) allocation to roots will increase and allocation to leaves will decrease; (3) leaves will show xeromorphic anatomical adjustments and (4) proline accumulation will be enhanced; (5) relative levels of mRNA of stress-related genes in leaves will be elevated. The additional, applied facet of our study was to test whether, and in which species, irrigation doses can be reduced below the established norm without compromising growth and affecting morphological, metabolic, and molecular characteristics of the seedlings.

## 2. Materials and Methods

### 2.1. Study Site

The experiment was conducted in the Rogoziniec forest nursery (52°18′ N 15°46′ E) located in Babimost Forest Division, SW Poland. The experiment was set up in an unheated 8 m × 50 m polyethylene tunnel.

*2.2. Plant Material*

*Fagus sylvatica* seeds and *Quercus petraea* acorns originated from the seed stands located close to Swiebodzin (52°15′ N, 15°32′ E), *Pinus sylvestris* seeds were from the local 'Rogoziniec' provenance (52°10′ N, 15°50′ E), and *Abies alba* seeds originated from the Sudety Foothills (51°7′ N, 15°55′ E), Poland.

*2.3. Experimental Design*

All seeds were subjected to stratification. At the end of May 2015 they were sown into individual cavities of 54-cavity polystyrene containers (Robin, France). Each cavity was filled with 430 mL of substrate consisting of peat and perlite (3:1 *v/v*, pH = 6.5) with the addition of 2 g of the 'Osmocote Exact Standard' slow-release fertilizer (N:P:K:Mg 15:9:12:2, with micronutrients). During a ten-day germination period containers were watered every day up to field capacity. Containers were placed on benches 0.30 m above the ground and arranged in a full-factorial completely randomized block design with species and irrigation levels as the fixed factors and blocks as the random factor. In our experiment, the irrigation norms recommended by Pierzgalski et al. [62] for forest nurseries in Poland were used as the benchmark. In these guidelines, the species of trees are classified into one of three groups: (1) conifers including *Pinus sylvestris* and *Abies alba* with one deciduous species *Tilia cordata*, (2) *Quercus*, *Fagus sylvatica* and *Tilia platyphyllos*, (3) other broadleaved deciduous species. From April to June, according to these norms, at the initial phase of growth, seedlings are irrigated every two days with 5 mm of water for *P. sylvestris* and *A. alba* and with 7 mm every three days for the broadleaved deciduous species. In our experiment, we relied on both the norms and nurseryman' experience. Four daily irrigation levels were established: 6 mm ($6 L/m^{-2}$, 100%), 4.5 mm ($4.5 L/m^{-2}$, 75%), 3 mm ($3 L/m^{-2}$, 50%), and 1.5 mm ($1.5 L/m^{-2}$, 25% of the maximal dose). When recalculated per area of one cavity of container, 12 (100%), 9 (75%), 6 (50%), or 3 mL (25%) of water were delivered every day. After leaf expansion, water amount reaching the substrate in cavity was reduced when compared to these values due to interception. To compare species-specific responses to the irrigation doses, we applied the same irrigation regime for each species. Each irrigation regime was replicated four times with four containers per species and irrigation treatment per block (4 blocks × 4 species × 4 treatments × 4 containers × 54 cavities). At the end of the experiment, the total number of seedlings was 8008. Seedlings were randomly selected, and the numbers of replicates per each analysis (*n*) are given in the Results. Additional containers with seedlings of Scots pine were placed around the experimental blocks to eliminate the edge effect.

*2.4. Microclimate*

Air temperature and relative humidity (*RH*) in the tunnel were monitored with two HOBO Pro v2 probes (OnSet Computer Corporation, Pocasset, MA, USA). The probes were attached at the height of 90 cm above the ground (close to the apical parts of seedlings) and logged data every twenty min. During the active growing phase from mid-June to the beginning of August mean air temperature ($T_{mean}$ ± SD, where SD is standard deviation) was 23.4 ± 7.4 °C and the average relative humidity ($RH_{mean}$ ± SD) was 61 ± 22%. In July $T_{mean}$ = 20.7 ± 6.0 °C, maximal temperature ($T_{max}$) was 38.8 °C, minimal temperature ($T_{min}$) was 10.8 °C, and $RH_{mean}$ = 65 ± 19%, $RH_{max}$ = 96%, and $RH_{min}$ = 28%. In August $T_{mean}$ = 24.2 ± 7.6 °C, $T_{max}$ = 42.5 °C, $T_{min}$ = 7.1 °C, and $RH_{mean}$ = 60 ± 23%, $RH_{max}$ = 96%, and $RH_{min}$ = 19%. Extremely high temperatures exceeding 40 °C occurred on 6 and 8 August between 2:00 and 4:30 p.m.

*2.5. Regulated Water Deficit*

The terms water deficit and drought have often been considered synonymous although water deficit is more commonly used when referring to water availability below field capacity, especially in relation to crop cultivation, whereas drought is generally related to low precipitation over a period

of time. In our study, seedlings were cultivated under controlled conditions in an unheated tunnel, therefore we use the term 'water deficit' for irrigation levels below the established norm.

Plants were watered by an overhead sprinkler system. The different irrigation levels were obtained by regulating the duration of water delivery over particular blocks. The water pressure (0.25 MPa in each sprinkler) and irrigation time were monitored using an automated system (Rathmakers, Germany). A pluviometer installed below the overhead sprinklers was used to measure water amount in millimeters in each irrigation treatment.

*2.6. Growth, Biomass Allocation, and Water Contents*

The height and root collar diameter of seedling were measured after 60 days of growth under experimental conditions ($n = 10$, $n$—number of seedlings per block, species and per treatment). Seedlings were subsequently removed from containers and their roots were gently washed. Plants were dissected into roots, stem and leaves and fresh mass (FM) of each fraction was determined. Biomass fractions were then dried at 65 °C for 7 days to constant weight using a climate cabinet (Pol-Eko, Wodzisław Śląski, Poland), and weighed for dry mass determination. Water amount in each type of organ was calculated by subtracting organ dry mass from its fresh mass and water content per unit of dry mass was obtained by dividing the water amount by dry mass of the organ fraction or by total seedling dry mass.

*2.7. Microscopic Observations*

On 13 August 2015 one mature, undamaged leaf per seedling was sampled from eight seedlings (two from each block) in 25%, 50%, and 100% irrigation regimes. To enable rapid sample processing for anatomical analysis we did not sample from the 75% irrigation treatment. A 2 mm wide segment from the central part of the needle or the broadleaf lamina (including midrib) was excised and fixed by vacuum infiltration for 2 h in a mixture of paraformaldehyde (2%) and glutaraldehyde (2%) in cacodylate buffer followed by overnight incubation under atmospheric pressure. Samples were dehydrated in graded ethanol series. *Abies*, *Fagus*, and *Quercus* samples were embedded in Technovit resin (Heraeus Kulzer GmbH, Hanau, Germany), sectioned to 5 μm with a microtome (Leica RM 2265, Leica Biosystems, Wetzlar, Germany) and stained with 0.01% solution of toluidine blue in 1% sodium borate. Sections of *Pinus* needles (35 μm thick) were obtained using a VT 1200 S vibratome (Leica Biosystems, Wetzlar, Germany). Micrographs were taken using a light microscope (Axioscope A1, Zeiss, Oberkochen, Germany) with an attached digital camera (AxioCam MRC5, Zeiss, Oberkochen, Germany). Micrometric traits were measured with Axiovision 4.9.1 (Zeiss, Oberkochen, Germany). In each leaf we determined the thickness of lamina, the length of palisade cells (mean of 3–5 cells per leaf, except in *Pinus* where palisade mesophyll does not occur), transverse area of xylem and phloem in the central vein, and mean and maximal diameter of conduits (based on 10 widest vessels in the angiosperms and tracheids in the conifer species).

*2.8. Determination of Leaf Proline*

Proline concentration was determined with the method of Bates et al. [63]. Fully expanded leaves from the apical portion of the seedlings were sampled on 7 August 2015. Purified proline was used to standardize the procedure for quantifying sample values. Acid-ninhydrin was prepared by warming 1.25 g ninhydrin in 30 mL glacial acetic acid and 20 mL 6 M phosphoric acid at 75 °C until dissolved. A 0.3 g sample of plant material was homogenized in 3 mL of 5% aqueous sulfosalicylic acid and the homogenate filtered through Whatman # 2 filter paper. A 2 mL aliquot of filtrate was incubated with 2 mL of acid-ninhidrin and 2 mL of glacial acetic acid in a test tube for 1 h at 100 °C, and the reaction was terminated in an ice bath. The reaction mixture was extracted with 4 mL toluene by mixing with a test tube stirrer for 15–20 sec. The chromophore containing toluene was aspirated from the aqueous phase, warmed to room temperature, and the absorbance read at $\lambda = 515$ nm (Spekol, CarlZeiss, Oberkochen, Germany) using toluene as blank. The proline concentration was determined

from a standard curve and calculated on a dry mass basis using the formula $S_k = K \times A \times T$, where: K is coefficient calculated with the standard curve, A is absorbance, and T is conversion factor accounting for the volume of toluene. Proline leaf concentration was expressed in $\mu g \ g^{-1}$ DM.

### 2.9. Expression of Stress-Related Genes

Leaf samples for RNA extraction were collected on 7 August 2015. Leaves without symptoms of necrosis or other visible damage were collected. In each species five seedlings per block and treatment were sampled and three samples were used for analyses. Leaf material was immediately frozen in liquid nitrogen and stored at −80 °C until RNA extraction.

Plant material was ground in liquid nitrogen and total RNA was extracted with 2% B-mercaptoethanol in the extraction buffer as described by Chang et al. [64]. Plant material was ground in liquid nitrogen and total RNA was extracted with 2% B-mercaptoethanol in the extraction buffer as described by Chang et al. [64]. The isolated RNA was dissolved in 30 μL ddH$_2$O. The extracted RNA was quantified using the Qubit®RNA BR Assay kit (Life Technologies, Carlsbad, CA, USA). The quality of the RNA was checked on a 1% agarose gel. One microgram of RNA per sample was treated with DNase RQ (Life Technologies) and used for cDNA synthesis (Transcriptor First Strand cDNA Synthesis Kit, Roche). The resulting cDNA was diluted 1:10 and used for quantitative real-time PCR in a LightCycler 96 (Roche, Manheim, Germany). To each well 2.5 μL of diluted cDNA, 10 μL of SYBR Green I Master (Roche, Manheim, Germany), 1.25 μL of 5 μM mixture of the forward and reverse primers and 5 μL of ddH$_2$O were added for the amplification reaction. Each biological sample for each gene was run twice. The PCR programs were as follows: 5 min at 95 °C, 45 cycles of amplification for 10 s at 95 °C, 10 s at primer annealing temperature (Supplementary Table S1), and 20 s at 72 °C. Specificity of PCR amplification was analyzed by melting curve and peaks (60–95 °C, with temperature ramp 2.2 °C per s) and verified on agarose gel.

After the RT-qPCR procedure was complete, the quantification cycles were determined using the second derivative maximum method implemented in the Roche LightCycler 96. The primer efficiency was determined by a dilution series and ranged between 1.8 and 2.0 for the genes of this study. The relative expression was analyzed with $\Delta\Delta C_T$ [65]. The potential reference genes were tested for their expression stability among the all treatment groups using Normfinder [66] implemented in the RefFindr [67]. Gene expression was normalized by the gene expression derived from housekeeping genes: Ubiquitin (*UBQ*, GenBank: AF461687) for *P. sylvestris,* Cyclophilin 2 (*CYC*, GenBank: ERP001867) for *Q. robur*, actin (*ACT*, GenBank: AM063027) for *F. sylvatica* and translation initiation factor IF-2 subunit alpha (*TIF2A,*) for *A. alba* (Supplementary Table S1). Real-time quantitative RT-PCR (qRT-PCR) was used to describe plants responses to water deficit. On the base of published data, we selected drought-related genes which were previously found by others to be related to drought stress response (up-regulated or down-regulated) or to abiotic stress in analyzed species. In *F. sylvatica*, abscisic acid (*ABA*)-related drought signaling genes were analyzed: *NCED1*-e9-cis-epoxy-dioxygenase-required for *ABA* biosynthesis (GenBank: DQ787262); *PP2C*-protein phosphate 2C-involved in *ABA* signal transduction (GenBank: AJ277743); *ERD10*-early responsive to dehydratation, an *ABA*-responsive transcription factor that attenuates *ABA* responses (GenBank: FR775803); stress protection *APX1*-ascorbate peroxidase genes encoding enzymes required for the scavenging of superoxide radicals, hydrogen peroxide and toxic aldehydes (GenBank: FR774767) [68]. In *Q. petraea* the following genes expression was investigated: α-tubulin (*TUB*) related with actively dividing tissues; Zn-finger protein of *Arabidopsis thaliana* (*ZAT11*)—a gene involved in oxidative stress-induced programmed cell death; glutathione-s-transferase (*GST*) and used primers were previously published by Makela et al. [69], however gene expression of *TUB* was omitted from further analysis due to efficiency of primers higher than 2.0. In *P. sylvestris*: genes involved in biosynthesis (*SPDS*-spermidine synthase, GenBank: HM236827) and metabolism of polyamines (PAO—polyamine oxidase, GenBank: HM236830; catalase (CAT, GenBank: EU513163), involved in reactive oxygen species homeostasis regulation; pyruvate decarboxylase (*PDC*) (GenBank: CO161777.1)

marker of alcohol fermentation in tissues under hypoxia; late-embryogenesis abundant protein (*LEA*), (GenBank: AAX68990.1) the well-known osmoprotectors; glutamate-cysteine ligase (*GCL*), (GenBank: AJ132540.1); glyceraldehyde-3-phosphate dehydrogenase (*GAPDH*) (GenBank: L07501) [70]. In *A. alba*: genes classified as early responsive to dehydratation (*ERD10*) (GenBank: P42759); chloroplastic beta-amylase 1, b isoform (*BAM1*.b)-(GenBank: Q9LIR6); two pore calcium channel protein 1 *TPC1* (GenBank: AT4G03560) involved in stomatal closure and abiotic stress response; xyloglucan endotransglucosylase/hydrolase (*XTH7*, GenBank: Q8LER3) related with cell growth; one of putative uncharacterized protein (*PUP1*.a10.13.9) (GenBank: A9NLY4) belongs to dehydrin family [59].

### *2.10. Statistical Analyses*

Prior to analyses, the data were tested for normality and homogeneity of variance in groups with Shapiro-Wilk's and Levene's test, respectively. Data were $\log_{10}$ transformed to fulfill the ANOVA conditions. Two-way ANOVA with species and irrigation regimes as fixed factors and blocks as the random factor was applied to test species, treatment and interaction effects. Results of two-way ANOVA are given in Supplement. Gene expression was examined with one-way ANOVA because different genes were analyzed for each species. One-way ANOVA with irrigation regimes as the fixed factor and blocks as the random factor was applied to compare the irrigation treatments within each species for organ water content, whole plant dry mass, biomass allocation, leaf proline concentration, anatomical leaf traits, and gene expression. The block effect was removed from ANOVA model when non-significant. The mean values for treatments within the species were compared with the analysis of contrasts. Significance level was $p < 0.05$.

## 3. Results

### *3.1. Seedling Water Status*

At time of harvest both gymnosperm species (*Abies* and *Pinus*) had higher LWC (leaf water content) and lower RWC (root water content) than the angiosperm species (*Fagus* and *Quercus*) whereas their SWC (stem water content) were similar (Table S1; Figure 1). Lower irrigation resulted in slightly reduced LWC in *Quercus* and *Pinus* although majority of pairwise contrasts within species were non-significant (Figure 1a). There were also slight decreases of RWC in *Fagus* and SWC in *Quercus* and *Abies* due to reduced irrigation (Figure 1b,c). Thus, the measurement of water contents indicated only moderate organ-level water stress.

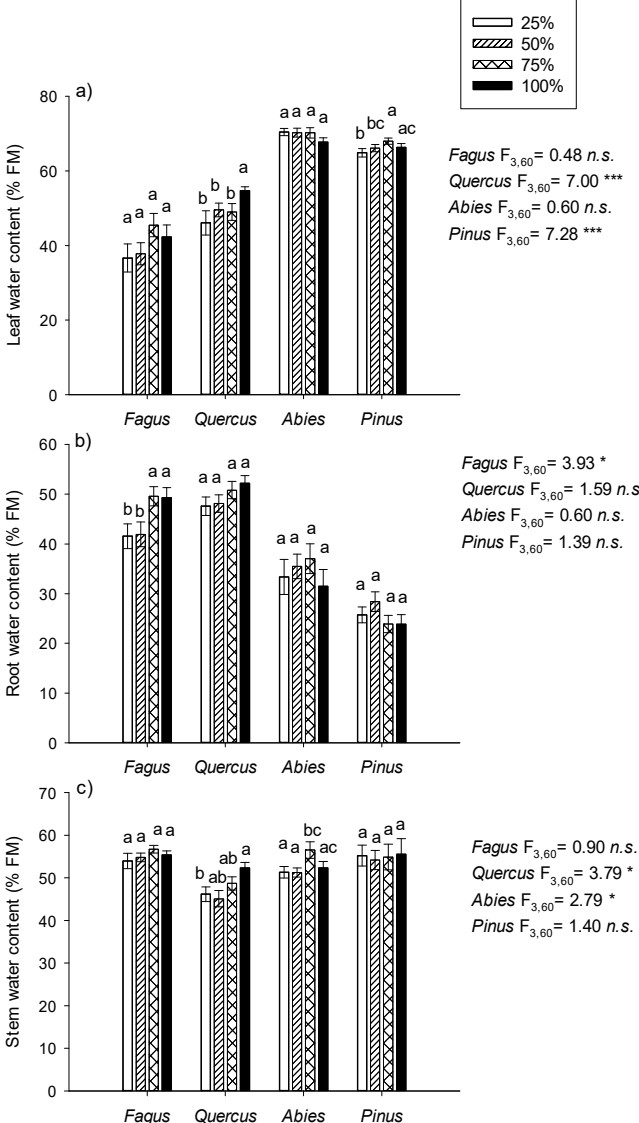

**Figure 1.** Water contents (determined as percentage of fresh mass) in leaves (LWC; (**a**)), roots (RWC, (**b**)) and stems (SWC; (**c**)) of *Fagus sylvatica*, *Quercus petraea*, *Abies alba*, and *Pinus sylvestris* seedlings grown at different irrigation treatments (100%, 75%, 50%, and 25% of the recommended dose). Bars indicate means ± SE (*n* = 16). One-way analysis of variance *F*-values for treatment effect within species and the associated *p* values are shown *** $p < 0.001$, * $p < 0.05$, n.s.—not significant). Means marked by the same letters (a, b, c) within species were not significantly different.

### 3.2. Growth

Reduction in water availability resulted in decreased height and root collar diameter of *Fagus*, *Quercus* and *Pinus* seedlings while in *Abies* the growth response to the different irrigation regimes was not significant (Table 1; Table S1). The magnitude of response was largest in *Fagus* (2.2 and 1.8-fold reduction in respectively, height and diameter), followed by *Quercus* (1.8 and 1.5-fold reduction and *Pinus* (1.5 and 1.2-fold). Significant height reductions occurred as irrigation was decreased from 100% to 75% (*Fagus* and *Quercus)* and from 75% to 50% and from 50% to 25% of the recommended dose (all three species). Mean slenderness indices (h/d) decreased with reduced irrigation in *Fagus, Quercus,* and *Pinus,* but not in *Abies* (Table 1).

**Table 1.** Effect of irrigation treatments (100%, 75%, 50% or 25% of the recommended dose) on height (h), diameter at root collar (d) and h/d ratio in *Fagus sylvatica*, *Quercus petraea*, *Abies alba*, and *Pinus sylvestris* seedlings. Means ± S.E. are given ($n = 40$). One-way analysis of variance *F*-values (with degrees of freedom for treatment effect and error term given in subscript) and the associated *p* values are shown separately for each species (*** $p < 0.001$, ** $p < 0.01$, * $p < 0.05$, n.s.—not significant). Means within column and within species marked by the same letters (a, b, c, d) were not significantly different.

| Species | Irrigation Treatment (% Recommended Dose) | h (mm) | d (mm) | h/d |
|---|---|---|---|---|
| *Fagus sylvatica* | 25 | 104 ± 5 [a] | 1.88 ± 0.08 [a] | 57 ± 2 [a] |
| | 50 | 124 ± 5 [b] | 2.13 ± 0.09 [b] | 59 ± 2 [a] |
| | 75 | 185 ± 7 [c] | 2.77 ± 0.08 [c] | 68 ± 3 [b] |
| | 100 | 225 ± 11 [d] | 3.37 ± 0.11 [d] | 67 ± 3 [b] |
| $F_{3, 156}$ | | 62.5 *** | 51.4 *** | 6.0 ** |
| *Quercus petraea* | 25 | 82 ± 3 [a] | 1.90 ± 0.07 [a] | 44 ± 2 [a] |
| | 50 | 94 ± 3 [b] | 1.97 ± 0.06 [a] | 48 ± 2 [ab] |
| | 75 | 117 ± 7 [c] | 2.48 ± 0.08 [b] | 48 ± 3 [ab] |
| | 100 | 148 ± 8 [d] | 2.78 ± 0.08 [c] | 54 ± 3 [b] |
| $F_{3, 156}$ | | 28.6 *** | 35.8 *** | 3.2 * |
| *Abies alba* | 25 | 35 ± 1 [a] | 1.00 ± 0.03 [a] | 36 ± 1 [a] |
| | 50 | 37 ± 1 [a] | 1.05 ± 0.03 [a] | 36 ± 1 [a] |
| | 75 | 38 ± 1 [a] | 1.03 ± 0.03 [a] | 38 ± 1 [a] |
| | 100 | 38 ± 1 [a] | 1.00 ± 0.02 [a] | 38 ± 1 [a] |
| $F_{3, 156}$ | | n.s. | n.s. | n.s. |
| *Pinus sylvestris* | 25 | 35 ± 1 [a] | 1.01 ± 0.03 [a] | 35 ± 1 [a] |
| | 50 | 39 ± 1 [b] | 1.06 ± 0.03 [a] | 37 ± 1 [a] |
| | 75 | 48 ± 1 [c] | 1.17 ± 0.04 [b] | 43 ± 1 [b] |
| | 100 | 51 ± 2 [c] | 1.18 ± 0.04 [b] | 45 ± 2 [b] |
| $F_{3, 156}$ | | 31.9 *** | 5.4 *** | 11.9 *** |

Biomass accumulation differed significantly among species (Table S1), with seedlings of the two angiosperms *Fagus* and *Quercus* reaching over 2.5 g dry mass under full irrigation, in contrast to the gymnosperms *Abies* and *Pinus* that reached only, respectively, 0.1 and 0.4 g (Figure 2a,b). Irrigation regime affected seedling growth in a dose-response manner in three out of the four species (Figure 2). Under 25% irrigation, dry mass of *Fagus* seedlings was over 5 times smaller than under the 100% dose, whereas in *Quercus* the difference was about three-fold (Figure 2a). There was, however, no effect of varied irrigation on the biomass of *Abies,* whereas in *Pinus* the biomass decrease between 100% and 25% doses was approximately by 1/3 (Figure 2b).

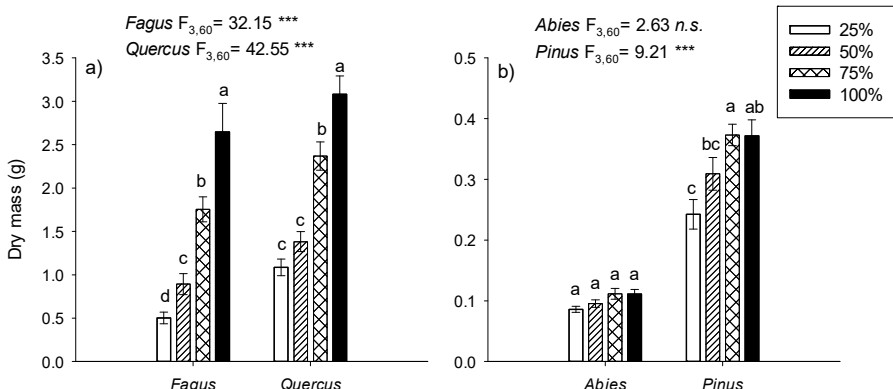

**Figure 2.** Total plant dry mass of (**a**) *Fagus sylvatica* and *Quercus petraea*, and (**b**) *Abies alba* and *Pinus sylvestris* seedlings grown at different irrigation treatments (100%, 75%, 50%, and 25% of the recommended dose). Bars indicate means ± SE (*n* = 16). Note the different scales. One-way analysis of variance *F*-values for treatment effect and the associated *p* values are shown separately for each species (*** $p < 0.001$, n.s.—not significant). Means marked by the same letters (a, b, c) within species were not significantly different.

### 3.3. Allocation

Alteration of biomass allocation in response to reduced irrigation was species-specific (Table S1; Figure 3a–d). The strongest response was observed in *Pinus*, in which ABMR (aboveground to belowground mass ratio) strongly declined, with irrigation reduction from 75 to 50%, resulting from an increased RMF (root mass fraction) and decreases in both LMF (leaf mass fraction) and SMF (stem mass fraction). In contrast, allocation in *Abies* was not affected by irrigation. In *Fagus*, reduced irrigation resulted in increased LMF and decreased SMF without effect on RMF or ABMR. Finally, in *Quercus* only a minor decrease in ABMR was noted at irrigation reduction from 100 to 75%, with variability in organ-level allocation not unaccounted for by irrigation.

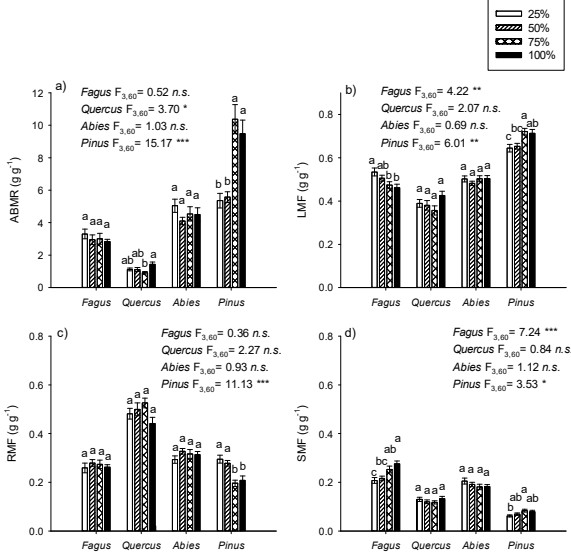

**Figure 3.** Patterns of biomass allocation in *Fagus sylvatica*, *Quercus petraea*, *Abies alba*, and *Pinus sylvestris* seedlings grown at different irrigation treatments (100%, 75%, 50%, and 25% of the recommended dose): aboveground to belowground mass ratio (ABMR; (**a**)), leaf mass fraction (LMF; (**b**)), root mass fraction (RMF; (**c**)) and stem mass fraction (SMF, (**d**)). Bars indicate means ± SE (*n* = 16). One-way analysis of variance *F*-values for treatment effect and the associated *p* values are shown separately for each species (*** $p < 0.001$, ** $p < 0.01$, * $p < 0.05$, n.s.—not significant). Means marked by the same letters (a, b, c) within species were not significantly different.

### 3.4. Leaf Proline Concentration and the Expression Levels of Selected Stress-Response Genes

Decline in irrigation resulted in an increase in proline concentration in the two angiosperms but not in the gymnosperm species (Figure 4). Specifically, the increased levels of proline were observed at 50 and 25% dose in *Fagus* and at 75% and less in *Quercus*. The magnitude of proline increase due to irrigation reduction was about two-fold in *Fagus* and somewhat less in *Quercus*.

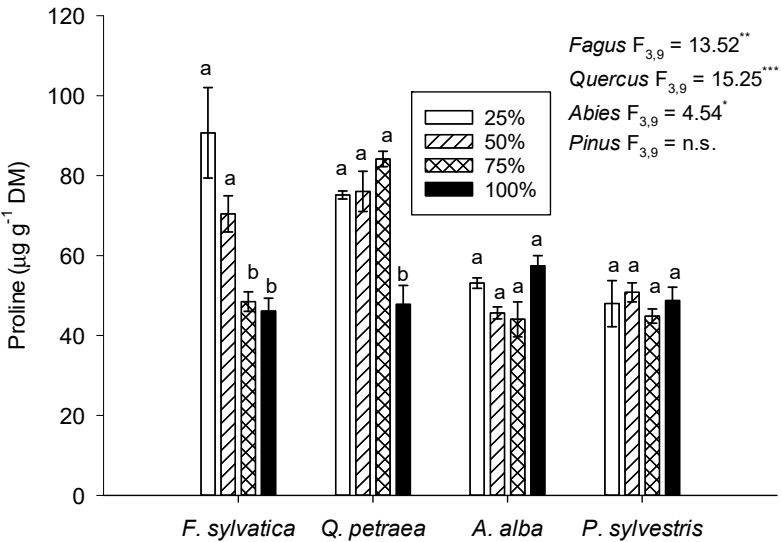

**Figure 4.** Leaf proline concentration of *Fagus sylvatica*, *Quercus petraea*, *Abies alba*, and *Pinus sylvestris* seedlings grown at different irrigation treatments (100%, 75%, 50% and 25% of the recommended dose). Bars indicate means ± SE ($n$ = 16). One-way analysis of variance *F*-values for treatment effect and the associated *P* values are shown separately for each species. Means marked by the same letters (a, b) within species were not significantly different as shown by preplanned contrasts. *P* symbols are: *** $p < 0.001$, ** $p < 0.01$, * $p < 0.05$, n.s. —not significant.

Gene expression showed very limited responses to a decrease in water dose. We observed an increase in relative gene expression of *ZAT11* in *Q. petraea* and in relative gene expression of *TPCC* in *A. alba* but saw no change in other tested transcripts (Supplementary Figure S1).

### 3.5. Anatomical Leaf Traits

Anatomical leaf traits were analyzed in seedlings exposed to three irrigation regimes: 25, 50, or 100% (Table S1; Figure 5). We found evidence for acclimation of some anatomical traits in *Fagus* and *Quercus*. Reduction of irrigation resulted in lower leaf lamina thickness in *Quercus*, and shortening of the palisade cells in *Fagus* and *Quercus* (with only a single palisade layer present; Figure 5A,B). Both angiosperms also showed reduced development of vascular tissues, with lower cross-sectional areas of xylem and phloem (Figure 5C,D), and their vessels exhibited smaller conduits (Figure 5E,F). The differences typically occurred between 100% and 50% but not between 50 and 25% irrigation levels. In contrast, none of the above traits varied among treatments in the two conifer species.

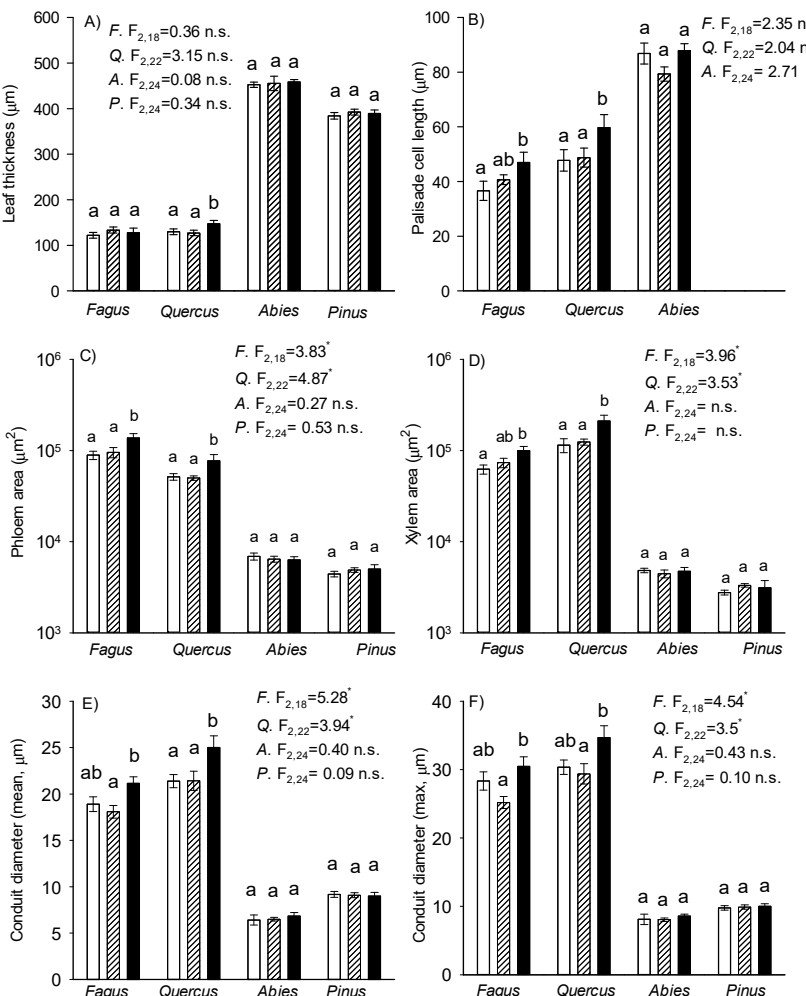

**Figure 5.** Anatomical parameters in leaves of *Fagus sylvatica* and *Quercus petraea*, and needles of *Abies alba* and *Pinus sylvestris* leaves in seedlings grown under different levels of irrigation (100%, 50% and 25% of the recommended levels). Bars represent the mean (±S.E.) (*n* = 8, *n*—number of seedlings per species and irrigation treatment). Anatomical features such as leaf thickness (**A**), palisade cell length (**B**), transverse midvein phloem (**C**) and xylem area (**D**), and within-leaf mean (**E**) and maximal (**F**) conduit diameter are shown. One-way analysis of variance *F*-values for treatment effect and the associated *p* values are shown separately for each species. *F*.—*Fagus*, *Q*.—*Quercus*, *A*.—*Abies*, *P*.—*Pinus*. Means marked by the same letters (a, b) within species were not significantly different. *P* symbols are: * *p* < 0.05, n.s.—not significant.

## 4. Discussion

### 4.1. Species-Specific Growth Responses to Irrigation

In this study of containerized-seedlings, large interspecific differences were observed in the response of seedling growth to reduced irrigation (Table S1). These differences were associated with the classification of the different species. In agreement with our first hypothesis, the broadleaf deciduous species, *Fagus sylvatica* and *Quercus petraea*, exhibited a greater magnitude of response to reduced irrigation than the two evergreen conifers *Abies alba* and *Pinus sylvestris*. Within the deciduous species, *F. sylvatica* was more impacted by reduced water availability than *Q. petraea*. The response of the deciduous species was in agreement with previous results obtained for seedlings and older trees of *Q. petraea*, indicating that it should be classified as a drought-tolerant species [71–73] and *F. sylvatica* as a drought-intolerant species [74,75]. Unexpectedly, the shade-tolerant and climax species, *A. alba*, exhibited a weak phenotypic response to reduced level of irrigation. On the other hand, the pioneer

and shade-intolerant species, *P. sylvestris*, exhibited a growth inhibition and a high morphological plasticity. The difference between the response of the studied conifers may be the result of their contrasting growth phenology as seedlings. The slow-growing species, *A. alba*, with its small amount of leaf biomass, requires less water than the faster growing, pioneer species *P. sylvestris*. Although all seedlings were exposed to drought during their rapid-growth phase, growth in *Abies* is naturally terminated sooner than in *Pinus*; thus the ability of *Abies* to respond to the decreased water availability may be restricted by its phenological cycle. This conservative growth pattern of *A. alba* may be a manifestation of its drought-avoidance strategy, exemplifying the trade-off between survival and growth [76]. Seeds of *A. alba* are also heavier than seeds of *P. sylvestris* and thus provide more resources to germinating seedlings during their initial growth. This advantage of *A. alba* seedlings may be especially important under water deficit conditions [77].

The interspecific differences observed in our experiment might have to some extent been induced by species-specific interception, which likely reduced the amount of water reaching the roots. In mature *P. sylvestris* stands in Germany, mean precipitation interception was 32% [78], and in leafed period in a *F. sylvatica* forest in Belgium, the level of interception was 28% [79]. These data cannot be directly compared with the interception of our study seedlings, but they highlight the importance of interception in the water balance of trees under natural conditions. In our study, however, the interception effect on water amount reaching the roots was substantially reduced because: (1) Water was delivered at high pressure: at the sprinklers the pressure was 4 bars, and at crowns of seedlings around 3.0–3.2 bars and never lower than 2.5 bars. Due to this high pressure, water reached the substrate in cavities even after full leaf expansion; (2) We cultivated our plants from seeds, thus at the beginning of the experiment, the interception was null and during germination and initial growth, it did not substantially affect the amount of water irrigating the substrate. An effect of interception was possibly important at the end of our experiment, after full leaf expansion. However, this effect of different interception might be at least partially compensated by the high pressure of irrigation water, high air humidity in the tunnel, and foliar water absorption which can be significant especially under water deficit [80].

## 4.2. Seedling Water Status

Relative tissue water content was used as a measure of plant water status. In our study, organ-level water content decreases caused by water deficit were small especially in the conifers, suggesting an efficient control of transpirational water loss. This was especially true in *Abies*. In addition, seedlings' water status could be substantially improved by foliar absorption of intercepted water, especially under water deficit. In an earlier study, a substantial improvement of water status, exceeding 1.0 MPa of water potential for drought-stressed *Juniperus monosperma*, was observed [80].

The two conifers showed much higher hydration of leaves across irrigation levels as compared to the two angiosperms. This may be attributed to the low amount of mechanical tissues and the large mesophyll content in juvenile needles. Interestingly, conifer root systems were much less hydrated than those of angiosperms, however, since conifer roots constituted a lower fraction of biomass than leaves, conifer seedlings held more water per gram total biomass than the angiosperm seedlings.

Our results suggest that the ability to accumulate and conserve foliar water may constitute a fundamental difference between drought-survival strategies of broadleaf deciduous and evergreen conifer species. This is supported, e.g., by a tighter stomatal control, substantial decrease in sap flow rates and transpiration in *P. sylvestris* compared with the more drought-tolerant *Quercus pubescens* [81,82]. Our suggestion is consistent with the higher growth rates of deciduous angiosperm seedlings achieved through a more intensive gas exchange, especially when expressed on the leaf mass basis. In the general scheme of ecological strategies, seedlings of both conifers thus appear to emphasize water conservation in contrast to specialization towards water acquisition and spending in seedlings of the broadleaf angiosperms.

### 4.3. Biomass Allocation

Evidence from individual and multi-species studies shows that plants typically respond to water deficit by increasing RMF and decreasing ABMR, LMF, and SMF [7,15,21,76]. This pattern was most closely represented by the response of *P. sylvestris*. In contrast, we did not observe modification of RMF in any other species while, surprisingly, *Fagus* responded to water deficit by increasing LMF. While the lack of modification of biomass allocation in *Abies* is understandable given its lack of growth response, the unexpected increase in LMF with drought observed in *Fagus* may reflect the opposite trend of SMF. Such a result may be explained by the intrinsic influence of plant size on allocation ratios: small plants tend to have low fraction of structural tissue (stem) and a large fraction of leaves; an effect often confounding the results of experimental treatments or environmental factors [21].

Increased RMR, as seen in droughted *Pinus*, is an important morphological adaptation to low availability of underground resources [21,83]. Under prolonged drought, moderate increases in root mass ratio may increase survival and ensure sufficient water supply for photosynthesis, whereas increasing this ratio beyond a level required to ensure survival and favorable water relations may impose construction and maintenance costs, resulting in net inhibition of plant growth. The role of biomass partitioning can be more important than either gas exchange or phenology as a mechanism of drought response as suggested, e.g., by a study using *Brassica rapa* [76].

A decrease in growth under water deficit may be advantageous for plants if it leads to a reduction of leaf transpiration area. However, under natural conditions, lower allocation to leaves and overall plant size reduction lowers the odds for individual's success in competition for light with other plants. This problem does not occur in container cultivation as long as seedlings are spaced to prevent significant competition. Additionally, higher RMR may be an advantageous trait for seedlings planted in dry sites. Given the prevalence of drought episodes, high seedling survival rates may be more important than their initial growth rates and biomass production [53,84]. Thus, morphological acclimation of seedlings to lower water doses in a nursery, especially the stimulation of increase in RMF, may benefit their survival after planting. However, this hypothesis should be verified by further experiments with seedlings of various species produced under different irrigation treatments in the container nursery and then planted and grown under natural conditions.

### 4.4. Osmoregulation and Alterations in Gene Expression in Response to Regulated Water Deficit

Plants synthesize saccharides, alcohols, and amino-acids including proline that accumulate within the protoplast, especially in the vacuole, leading to a decrease in cell osmotic potential [30,55,85]. Besides facilitating soil water extraction, some of these compounds, including proline, perform protective roles against oxidative damage associated with reduced gas exchange [86]. Although proline accumulates in response to water deficit in both angiosperm trees [87] and conifers [88], in our experiment such increase in proline concentration occurred only in angiosperm species but not in the conifers. This result, together with the relative homeostasis of needle water content in conifers, suggests that needle-level signal of water deficit was not sufficient to induce this biochemical response. Alternatively, conifers might accumulate some alternative osmolytes [89]. The utility of proline as an indicator of water deficit stress, although widely accepted [90], certainly depends on the species, ecotype, and stress intensity as well as tissue type [87,89].

The relative abundance of most of the analyzed drought-response-related transcripts was not affected by the different levels of irrigation except for *ZAT11* in *Q. petraea* and *TPC1* in *A. alba* seedlings (Supplementary Figure S1). *ZAT11* was involved in oxidative stress-induced programmed cell death [91] and positive regulation of primary root growth in *A. thaliana* [92]. Higher *ZAT11* could be associated with a higher production of reactive oxygen species due to drought stress. An increase in *TPC1* was involved in stomatal closure and abiotic stress responses in *Triticum aestivum* [93]. *TPC1* was proposed as a reference gene with stable expression for drought stress [59], but we found the opposite result. Under a higher water deficit than that in our study, *Populus alba* × *Populus tremula* var. *glandulosa* increased transcript levels of several drought markers in leaves and roots [94]. In agreement with our

results, Muilu-Mäkelä et al. [70] reported that the expression of genes involved in biosynthesis was not affected by drought stress in *P. sylvestris* seedlings. A wider screening of potential marker genes would certainly increase the likelihood of detecting stress at the level of gene expression.

*4.5. Leaf Anatomy*

Modifications of anatomical leaf traits in response to reduced irrigation were rather minor and involved small reduction in lamina thickness in *Quercus*, decrease in palisade cell length in *Quercus* and *Fagus* and decreases in the amounts of xylem and phloem tissues as well as in vessel diameters in these two angiosperms. In contrast, none of the conifer needles showed significant modifications. The character of changes in the two broadleaved species was only partly consistent with the previously described modifications induced by drought. Leaves that express constitutive or induced xeromorphism are often thicker than mesomorphic leaves and show greater development of palisade tissue [23,95,96]. Reduction, rather than increase, in leaf thickness and palisade cell length observed in our study suggests the occurrence of a leaf-level water deficit during leaf expansion and may be classified as symptom of water stress rather than adaptive modifications. On the other hand, the decrease in transverse areas of xylem and phloem are consistent with acclimation to lower gas exchange. Such plastic changes have been reported in both drought and salinity stressed plants [97,98]. Changes in leaf conduit diameter have rarely been examined in studies of drought acclimation, although the few studies reporting such data also show that vessels or tracheids become narrower in leaves developing under drought [88,99]. Such modification may reflect reduced demand for water transport capacity under decreased stomatal conductance and, as an additional benefit, reduce the vulnerability of vessels to embolization by decreasing the occurrence of 'rare leaky pits' [100].

The quantitatively small responses of angiosperm leaves and lack of response of gymnosperm needles to drought suggest that water deficit in meristematic and growing shoot tissues at the time of leaf development was not severe enough to induce significant plasticity of these traits (see Ivancich et al. [101] and Binks et al. [24] for similarly conservative responses). While leaves of young seedlings are known to be anatomically less plastic than those of later ontogenetic stages [102], it is also likely that the adjustment of growth rate and/or allocation of biomass, coupled with physiological adjustment, helped to at least partially offset the effect of lower availability of water and preserve the plant water status at least during the period of leaf expansion.

## 5. Conclusions

The divergent responses of tree species to regulated water deficit indicate that the irrigation regimes in container nurseries would benefit from species-specific adjustment according to water requirements. Our results indicate that in case of evergreen conifer species the quantity of irrigation water could be reduced by 25% (for *P. sylvestris*) to 75% (*A. alba*), and in case of deciduous broadleaf species (*F. sylvatica* and *Q. petraea*) by 25% relative to the full dose of 6 mm without significantly affecting growth and inducing morphological and physiological mechanisms of defense against water deficit. Although in both angiosperms growth was sensitive to water deficit, only very modest induction of stress indicators was observed, suggesting reduction of water dose could be considered as part of a field-hardening routine. This recommendation, however, should be validated by further experiments with seedlings of various species produced under the different irrigation regimes in a container nursery and then planted and grown under natural conditions.

**Supplementary Materials:** The following are available online at http://www.mdpi.com/1999-4907/11/9/1011/s1, Figure S1: Relative transcript abundance of selected response-to-abiotic-stress genes in leaves of *Fagus sylvatica*, *Quercus petraea*, and needles of *Abies alba* and *Pinus sylvestris* seedlings grown under different levels of irrigation (100%, 75%, 50% and 25% of the recommended level), Table S1: Results of a two-way ANOVA for growth parameters, biomass allocation, proline leaf concentration and leaf anatomical traits for *Fagus sylvatica*, *Quercus petraea*, *Abies alba* and *Pinus sylvestris* seedlings grown at different irrigation treatments (100%, 75%, 50% and 25% of the recommended dose) in a container nursery.

**Author Contributions:** Study conception and design: W.B., P.R., W.K., acquisition of data: P.R., W.K., T.P.W., E.P.-K., B.P., A.J.; analysis and interpretation of data P.R., T.P.W., E.P.-K.; drafting of first version of the manuscript: P.R. and final version: P.R., T.P.W., E.P.-K., B.P., W.B., W.K. and A.J. All authors have read and agreed to the published version of the manuscript.

**Funding:** This research was funded by The State Forests National Forest Holding and The APC was funded by Ministry of Science and Higher Education grant "Regional Initiative Excellence" in years 2019–2022, No. 005/RID/2018/19.

**Acknowledgments:** We thank M. Wisniewski for help in maintaining of the experiment.

**Conflicts of Interest:** The authors declare no conflict of interest.

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
