# Peer review of "Practical Implications of Different Phenotypic and Molecular Responses of Evergreen Conifer and Broadleaf Deciduous Forest Tree Species to Regulated Water Deficit in a Container Nursery"

_forests, doi:10.3390/f11091011_

Round 1

Reviewer 1 Report

Specific comments

P2 Line 95 – 96 Where does the ranking of drought tolerance of the study species come from? Please cite studies.

P13 Line 467 – 469 There seems to be little evidence to support the statement regarding the expression results suggesting that water deficit stress was efficiently moderated by stomatal closure. I suggest the authors moderate this statement as robust conclusions cannot be drawn from this analysis using a small number of candidate genes in my opinion.

General comments

Overall, the wiriting style is clear and concise.

In general, it would be nice to see measurements of leaf water potential in a sub-set of each species at each level of irrigation to quantify the level of water deficit/stress experienced. From a theoretical perspective, it would also be interesting to study the same growth responses and ‘markers of water stress’ under greater water stress than was imposed in this experiment to increase the interest beyond a study focussed on practical outcomes.

As per my specific comment above, choosing a small number of candidate genes in each species to study differential expression in the different treatments limits the inference space of the findings compared with studying relative abundance of transcripts genome-wide, for example, and this should be acknowledged in the manuscript.

It would be interesting to examine intraspecific variation in the traits (growth responses and ‘markers of water stress’) studied to indicate if there is potential to select for more drought tolerant genotypes within species.

Reviewer 2 Report

Review Forest 910303

Practical Implications of Different Phenotypic and Molecular Responses of 2 Evergreen Conifer and Broadleaf Deciduous Forest Tree Species to Regulated 3  Water Deficit in A Container Nursery

The manuscript aimed at evaluating the drought adaptability of juvenile trees grown in container nursery: 2 evergreen conifers (Abies alba, Pinus sylvestris) and 2 broadleaf deciduous forest trees (Fagus sylvatica, Quercus petraea). Plant response to various irrigation treatments (100%, 75%, 50%, 25%) was evaluated using morphological, anatomical, metabolic and molecular variables.  Conclusion was assessed about the specific response of each species, and/or functional group level to water deficit, and recommendations were provided to reduce water irrigation for seedlings.

General comments:

Experimental design rely on established norms to grow seedling in Central European conditions, under polyethylene tunnel and water delivering with automatic systems. General information about species water requirements and drought vulnerability ranking were defined according to sapling or adult trees. Therefore, the objective of the study was to evaluate response of tree seedling from 2 functional  group level to reduced irrigation, and provide recommendation for water saving.

The experimental design and statistical analysis were well designed to address the question. Experiments were clearly described in Material and method. In the results part, the metabolic response of the 4 seedling trees submitted to different irrigations is well described. My major comments are about gene expression results, discussion and conclusion.

  • Gene expression study was performed to evaluate the stress response at the transcriptional level. On the eighteen genes analyzed, selected according to former studies, only two varied significantly according to water deficit. Physiological measurements showed that your plants responded to water stress, which was confirmed by proline content. So you have undoubtedly induced a transcriptional stress response. You were just unlucky in your gene selection. If you wish to keep these results in your publication, I suggest to transfer them (table or graph) in supplementary data, and do not discuss them in the Results part. You can comment about the 2 significant genes in the discussion. However, you cannot make any assumptions using these results.
  • The whole discussion does not clearly rely on the results. Thus, it is difficult to support conclusions which seemed highly speculative. Discussion must be rewrite carefully.
  • In the conclusion, you propose water reduction specific to species. These recommendations are valuable but validation must be performed, as for example growth monitoring on growing trees and on the field.

Specific comments:

  • Age of seedlings is more useful than date of experiments
  • I do not understand why one way anova was applied for some analysis instead two way anova
  • 4: You performed a two way anova to highlight difference in leaf proline concentration between species, treatments and interaction of both. The results are not well explained, and are confusing. Line 364, the result correspond to a one way anova to estimate the difference between species for 100% irrigation only. On the graph, you give the significance for the 3 factors using a two way anova. Then, the letters you assigned on the bars for each species are the results of the mean comparison inside species (from one way anova or treatment effect from two way anova?). Considering your objective, comparison of different variables between species, a two way anova is more adapted. On the graph, you should indicate that the mean comparison is for the treatment inside species. In sup data, you can provide a table with the mean comparison for species, treatment and interaction.
  • 5: Annotations about species are lacking in graphs 5b, c, d, e. Annotation of F test result must be homogenized between graphs. About letters, same comment than for Fig.4.
  • 6: TUB was not analyzed because of technical raisons so you do not have to mention it in this article, except in the discussion if you want to explain why you have only 2 genes for Quercus. Statistical results described line 399 are different than the ones on the Fig.6. Statistical results of non significant GST are not useful. P-values are provided in the Fig.6 unlike other graphs. From a general point of view, I suggest to avoid presenting the results of gene expression (see my comment above).

Round 2

Reviewer 2 Report

The manuscript has much improved over the previous version.